# Comprehensive review of safety in Experimental Human Pneumococcal Challenge

Ryan E. Robinson[1,2], Christopher Myerscough[1], Nengjie He[1,3], Helen Hill[1], Wendi A. Shepherd[4], Patricia Gonzalez-Dias[1], Konstantinos Liatsikos[1], Samuel Latham[1], Fred Fyles[1], Klara Doherty[1,5], Phoebe Hazenberg[1], Fathimath Shiham[1], Daniella Mclenghan[1], Hugh Adler[1,2], Vicki Randles[1,2], Seher Zaidi[1,2], Angela Hyder-Wright[1,6], Elena Mitsi[1], Hassan Burhan[1,2], Ben Morton[1], Jamie Rylance[1,7], Maia Lesosky[1,3], Stephen B. Gordon[1,7], Andrea M. Collins[1,2☯], Daniela M. Ferreira[1,8☯]*

1 Clinical Sciences Department, Liverpool School of Tropical Medicine, Liverpool, United Kingdom, 2 Respiratory Research Group, Liverpool University Hospitals Foundation Trust, Liverpool, United Kingdom, 3 Global Health Trials Unit, Liverpool School of Tropical Medicine, Liverpool, United Kingdom, 4 North West Health Protection Team, UK Health Security Agency, Liverpool, United Kingdom, 5 University of Liverpool, Liverpool, United Kingdom, 6 Clinical Research Network, Liverpool, United Kingdom, 7 Malawi Liverpool Wellcome Research Programme, Blantyre, Malawi, 8 Oxford Vaccine Group, University of Oxford, Oxford, United Kingdom

☯ These authors contributed equally to this work.
* Daniela.Ferreira@paediatrics.ox.ac.uk

**Data Availability Statement:** All relevant data are within the paper and its Supporting Information files.

## Abstract

### Introduction

Experimental Human Pneumococcal Challenge (EHPC) involves the controlled exposure of adults to a specific antibiotic-sensitive *Streptococcus pneumoniae* serotype, to induce naso-pharyngeal colonisation for the purpose of vaccine research. The aims are to review comprehensively the safety profile of EHPC, explore the association between pneumococcal colonisation and frequency of safety review and describe the medical intervention required to undertake such studies.

### Methods

A single-centre review of all EHPC studies performed 2011–2021. All recorded serious adverse events (SAE) in eligible studies are reported. An unblinded meta-analysis of col-lated anonymised individual patient data from eligible EHPC studies was undertaken to assess the association between experimental pneumococcal colonisation and the frequency of safety events following inoculation.

### Results

In 1416 individuals (median age 21, IQR 20–25), 1663 experimental pneumococcal inoculations were performed. No pneumococcal-related SAE have occurred. 214 safety review events were identified with 182 (12.85%) participants presenting with symptoms potentially in keeping with pneumococcal infection, predominantly in pneumococcal colonised

**Funding:** The author(s) received no specific funding for this work.

**Competing interests:** The authors have declared that no competing interests exist.

individuals (colonised = 96/658, non-colonised = 86/1005, OR 1.81 (95% CI 1.28–2.56, $P$ = <0.001). The majority were mild (pneumococcal group = 72.7% [120/165 reported symptoms], non-pneumococcal = 86.7% [124/143 reported symptoms]). 1.6% (23/1416) required antibiotics for safety.

## Discussion

No SAEs were identified directly relating to pneumococcal inoculation. Safety review for symptoms was infrequent but occurred more in experimentally colonised participants. Most symptoms were mild and resolved with conservative management. A small minority required antibiotics, notably those serotype 3 inoculated.

## Conclusion

Outpatient human pneumococcal challenge can be conducted safely with appropriate levels of safety monitoring procedures in place.

## Introduction

Controlled Human Infection Models (CHIM), in which individuals are exposed to pathogens of interest, provide a methodology for studying disease pathogenesis, understanding immunological correlates of protection, and assessing vaccine candidates' therapeutic efficacy. Currently, some regulators do not consider experimental challenge a medicinal product [1]; and therefore CHIMs do not necessarily have the same level of regulatory scrutiny as a Clinical Trial involving an Investigational Medicinal Product (CTIMP). However, the World Health Organisation (WHO) states that as these studies may evaluate the efficacy of future vaccines they should be treated similarly regarding regulations [2], in addition to Good Clinical Practice (GCP) compliance [3]. Given the risks inherent in replicating natural carriage/infection with a known pathogen, meticulous safety monitoring systems are crucial when protecting participants from harm.

In some CHIM studies participants are pre-emptively hospitalised for monitoring. This precaution is due to the challenge pathogen posing a potentially high level of risk to the participants' health if unmonitored and untreated, alongside the potential transmission risk to the public. Rigorous participant selection, education and early engagement with local health practitioners may mitigate this risk [4]. In other CHIMs, with asymptomatic colonisation, rather than disease as the aim, outpatient monitoring of well-informed participants may suffice.

The Experimental Human Pneumococcal Challenge (EHPC), a CHIM first developed in 2001 by McCool et al [5,6], has been used [7] to understand pneumococcal colonisation biology and links with transmission, correlates of protection and to accelerate pneumococcal vaccine development. Participants are experimentally inoculated with *Streptococcus pneumoniae* (pneumococcus), to explore immune responses mimicking "natural" exposure [7]. In this model, prevention of nasopharyngeal pneumococcal carriage acts as a surrogate for vaccine-induced immunity when testing vaccine efficacy [8], significantly reducing the required sample size and study duration. Previous research has demonstrated experimental colonisation with serotype 6B (SPN6B) to be asymptomatic in adults [9]; however other observational studies have reported mild symptoms in children during natural colonisation [10]. An increased frequency of upper respiratory tract (URT) symptoms was observed for serotype 3 [11], suggesting that serotype may impact the development of symptoms during colonisation.

CHIMs are considered safe for determining the efficacy of clinical interventions; however this has not been fully evidenced within pneumococcal CHIM through large-scale, objective analyses. Our aims are, therefore,: 1) A comprehensive description of all adverse events (AE, serious and non-serious) in EHPC at our centre 2) An assessment of AE severity (and hence re-consideration of the appropriateness of outpatient management) 3) An unblinded comparison of pneumococcal colonised/non-colonised experience of symptoms, including consideration of gender and other pneumococcal disease risk factors 4) Exploration of the influence of inoculum serotype on symptoms and 5) A review of safety processes and re-consideration of the level of precaution/effectiveness of interventions, to help inform other CHIM programmes.

We hypothesise that an outpatient pneumococcal challenge can be performed safely with appropriate monitoring, and that the pneumococcal colonisation status, challenge inoculation method and the administered serotype may influence the frequency of safety review required post-inoculation.

## Methods

A single-centre review of safety data from EHPC studies at our centre between January 2011–June 2021, using an individual patient data meta-analysis approach for collated data. Inclusion criteria were: 1) Human participants 2) Outcome of nasopharyngeal pneumococcal colonisation, 3) Reported original results, and 4) Complete safety data available for review. Studies without extractable data were excluded. We accessed historical safety records and anonymised participant case report forms (CRFs) from published and unpublished studies. Data extraction was independently performed by 5 reviewers using a standardised data extraction form, with discrepancies resolved by group consensus.

All included studies had approval from a UK Research Ethics Committee (NHS Health Research Authority REC North West–Liverpool Central, REC IDs included in Table 1) and had been prospectively registered on a clinical trials database where required. Prospective approval for this service evaluation was obtained from the Liverpool University Hospitals Trust Audit Department (ID 10354). All participants provided written informed consent, and none had previously received a pneumococcal vaccine. Safety was defined as the absence of Serious Adverse Events (SAEs) directly related to experimental pneumococcal inoculation, as determined by an independent Data Safety and Monitoring Committee (DSMC).

An overview of EHPC study timepoints and procedures is illustrated in Fig 1A and the wider safety monitoring processes pre-and post-inoculation are summarised in Fig 1B. Inclusion and exclusion criteria utilised across all EHPC studies are detailed in S1 Table.

### Participant symptom reporting and review

All participants were asked about the presence of symptoms at every visit, and if reported, underwent medical review by a study clinician. Symptomology, onset, severity, and duration were documented in the CRF. Further information on participant symptom assessment and investigation is detailed in the S1 File. Of note, study clinicians were unblinded to colonisation status throughout, impacting clinical suspicion during review. Participants were routinely blinded to their colonisation status until the end of follow-up.

### Identification of participant episodes requiring safety review post-experimental challenge

All EHPC studies generate a weekly detailed safety report containing clinically relevant during the study period. Identified studies were reviewed in a two-stage process:

**Table 1. Experimental Human Pneumococcal Challenge studies performed between 2011–2021.**

| Full Study Title | Short title | Year | Study type | CTIMP | REC ID | Sample size | Mean age years (SD) | NHV# | Serotype | Inoculation | Inoculation method | Doses (CFU) x10⁴ | Study Follow up (days) | Monitoring period (days) | SAE |
|---|---|---|---|---|---|---|---|---|---|---|---|---|---|---|---|
| EHPC: Establishing SPN3 Challenge Model [11] | Pneumo 1 | 2019 | Cohort | No | 19/NW/0238 | 96 | 22.6 (4.5) | Yes | SPN3 | 1 | pipette | 10–160 | 14 | 3 | No |
| Lifebuoy Bar Soap Handwashing Randomised Control Trial | Unilever Lifebuoy Soap | 2019 | RCT | No | 19/NW/0043 | 156 | 25.3 (6.5) | Yes | SPN6B | 1 | hand | 3200 | 10 | 3 | No |
| Streptococcus pneumoniae Nasopharyngeal Experimental challenge study of Attenuated Strains | SNEAS | 2018 | RCT | No | 18/NW/0481 | 146 | 23.7 (7.0) | Yes | 6B/6B GMO | 1, 2 or 3 | pipette | 80 | 196 | 3 | No |
| EHPC: The effect of asthma on immune response to Pneumococcus [12] | Asthma | 2016 | Cohort | No | 16/NW/0124 | 50 | 24.8 (7.7) | No, asthma | SPN6B | 1 or 2 | pipette | 80 | 29 | 7 | No |
| EHPC: The effect of new strains (types) of bacteria in healthy participants | New Strains | 2016 | | No | 15/NW/0931 | 16 (23F) | 26 (7.7) | Yes | SPN23F, SPN15B | 1 | pipette | 20–160 | 14 | 7 | YES (1) |
| | | | Cohort | | | 54 (15B) | | | | 1 or 2 | | 80 | | | |
| EHPC: The effect of age on immune function[13] | Ages | 2016 | Cohort | No | 16/NW/0031 | 64 | 63.5 (7.2) | Yes, older adults | SPN6B | 1 or 2 | pipette | 80 | 29 | 7 | No |
| Hand to nose transmission of streptococcus pneumoniae in healthy participants–pilot study [14] | Hand to nose Pilot study | 2017 | Cohort | No | 17/NW/0054 | 63 | 22.6 (4.9) | Yes | SPN6B | 1 | hand | 3200 | 9 | 7 | No |
| Early and late nasal and tonsil cell responses during human pneumococcal colonisation | Nose and Throat | 2016 | Cohort | No | 17/NW/0029 | 43 | 26.3 (8.3) | Yes | SPN6B | 1 | pipette | 80 | 23 | 7 | No |
| Hand to nose transmission of Streptococcus pneumoniae in healthy participants; randomised control trial assessing the effect of hand washing on transmission | Handwash Study | 2017 | RCT | No | 17/NW/0658 | 136 | 21.81 (5.1) | Yes | SPN6B | 1 | hand | 3200 | 10 | 3 | No |
| The Effect of Live Attenuated Influenza Vaccine on EHPC 2 [15] | LAIV study 2 | 2016 | RCT | Yes | 14/NW/1460 | 195 | 21.0 (3.3) | Yes | SPN6B | 1 | pipette | 80 | 27 | 7 | YES (1) |
| The effect of mucosal sampling on experimental human pneumococcal colonisation, a pilot study [16] | Mucosal Study | 2015 | Cohort | No | 15/NW/0146 | 20 | 28.3 (7.4) | Yes | SPN6B | 1 | pipette | 80 | 14 | 7 | No |

(Continued)

Table 1. (Continued)

| Full Study Title | Short title | Year | Study type | CTIMP | REC ID | Sample size | Mean age years (SD) | NHV[#] | Serotype | Inoculation | Inoculation method | Doses (CFU) x10⁴ | Study Follow up (days) | Monitoring period (days) | SAE |
|---|---|---|---|---|---|---|---|---|---|---|---|---|---|---|---|
| The Effect of Live Attenuated Influenza Vaccine on EHPC 1 [15] | LAIV study 1 | 2015 | RCT | Yes | 14/NW/1460 | 130 | 21.0 (4.2) | Yes | SPN6B | 1 | pipette | 80 | 29 | 7 | No |
| Pneumococcal Conjugate Vaccine-13 (Prevenar-13) and EHPC Study [8] | PCV EHPC | 2013 | RCT | Yes | 12/NW/0873 | 96 | 23.8 (6.7) | Yes | SPN6B | 1 | pipette | 80 | 21 | 7 | YES (1) |
| EHPC: Dose ranging and reproducibility [17] | Gates EHPC | 2011 | Cohort | No | 11/NW/0592 | 151 | 22.7 (6.0) | Yes | SPN23F, SPN6B | 1 | pipette | 10–320 | 42 | 7 | No |

# Normal healthy volunteers.

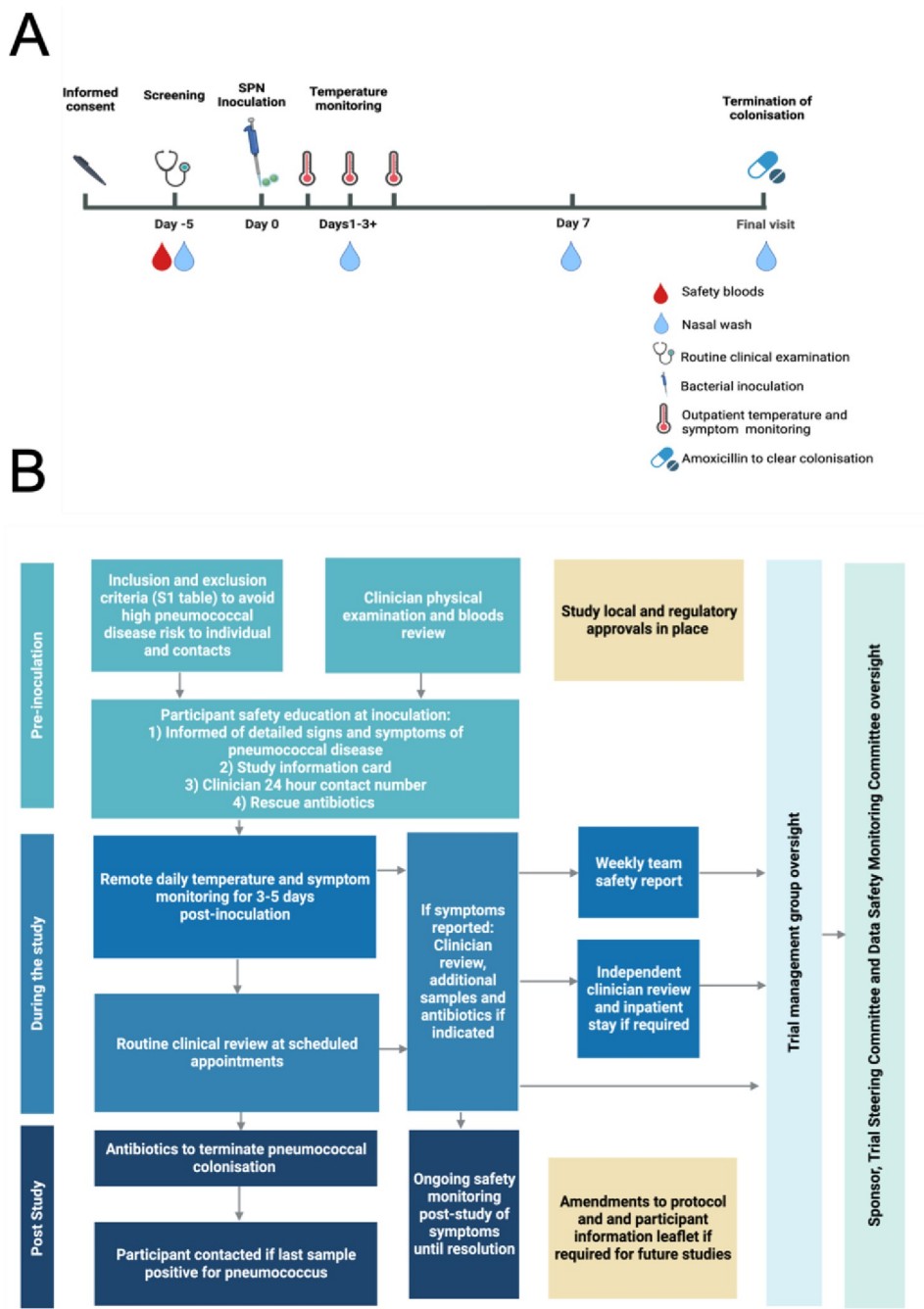

**Fig 1. EHPC study design and safety procedures.** A) A schematic demonstrating EHPC study timepoints and procedures. Interventions including vaccination or further pneumococcal inoculation can occur at pre-specified study visits. Experimental Colonisation status is identified in nasal wash samples taken at minimum of day 2 and day 7 post-inoculation. Participants are monitored for symptoms from inoculation until their final visit. **B)** Overview of safety procedures in EHPC. A schematic diagram summarising the safety processes in place prior to participant inclusion in the study, then following inoculation and post-study. Regulatory processes are not included in this diagram.

1. Study safety report review: Weekly safety reports for each study were reviewed for clinically relevant information on reported symptoms, extra clinical reviews, additional investigations, and all medically-significant occurrences. Potential clinical interactions were identified from these reports for further review.

2. Detailed CRF review: CRFs of relevant participants identified in the study safety report were reviewed by an experienced clinician to identify demographic information, study, challenge serotype and dose, inoculation method, colonisation status, and where safety incidents have occurred, alongside other key information pertaining to this.

Some EHPC studies involved multiple experimental challenges (with different serotypes and doses). Each challenge episode was treated as an individual event. All clinical episodes were reported separately unless deemed a continuation of the same symptom event. The follow-up period was considered closed at the last study visit. Data was collated in a purpose-built Microsoft© Access database.

## Defining SAE and Adverse Events (AE)

Definitions for key safety reporting terminology as defined in the Medicines and Healthcare products Regulatory Agency (MHRA) 'GCP guide' [3] were utilised, depending on the study protocol to standardise methods across CTIMP and non-CTIMP studies. These definitions were implemented as judged by consensus of an experienced clinical team and DSMC for SAE.

## Defining potential pneumococcal related symptoms post-experimental challenge

*S. pneumoniae* is responsible for a broad spectrum of disease, meaning defining any presenting symptoms potentially related to pneumococcus is challenging. The UK Health Security Agency defines pneumococcal disease as 'invasive' and 'non-invasive', with five main clinical syndromes: meningitis, pneumonia, septicaemia (all invasive), otitis media and sinusitis (non-invasive) [18]. This study, therefore, focused on the symptomology of these disease areas. A literature review identified the clinical presentation and features of immunocompetent adults with these clinical syndromes due to pneumococcal infection (search terms detailed in S2 Table and results in S3 Table). This list was non-exhaustive, and clinician discretion was applied when assessing symptoms, with reviewer group consensus where a disagreement occurred. Medical events deemed non-related to pneumococcus were collected to ensure unexpected patterns of pneumococcal disease were not overlooked in the analysis.

## Grading of participant symptom severity post-experimental challenge

No universally adopted system exists for grading AEs in CHIMs. To allow comparison the grading system (S4 Table, adapted from the Food and Drug Administration (FDA)/Center for Biologics Evaluation and Research, 'Grading of AE in healthy volunteers in vaccine trials' [19] was retrospectively applied by an independent clinician. If a participant received antibiotics or medical intervention, symptoms were deemed grade 2 or above.

## Statistical analysis

Demographic data, colonisation data, and frequency of review were reported using descriptive statistics. The frequency, safety event type and symptoms in colonised and non-colonised individuals were compared using Fisher's Exact Test. An individual patient data meta-analysis

methodology was utilised to estimate the association between pneumococcal colonisation and symptoms ('all reported' and 'potentially pneumococcal related') requiring safety review. Two-stage meta-analysis using unadjusted binomial mixed effects models with a logit link and study ID as a random effect was used to estimate unadjusted associations (reported as odds ratio (OR), 95% confidence intervals [CI]) overall and by strata for visualisation in forest plots. Inverse variance weights and Hartung-Knapp adjustment were used for this model. While the interpretation of forest plots is limited as the data relates to a single centre, this method remains an effective means of visually displaying this data. A single-stage, co-variate adjusted, individual patient meta-analysis methodology was also utilised, again under a binomial mixed effects model with logit link, fitting the study as a random effect to identify an association with co-variates that may adjust or confound the association of interest. Sensitivity analyses were performed to assess the association with challenge serotype and inoculation method. All *P*-values were two-tailed and considered significant if *P* <0.05. All analyses were performed in R (version 4.2.0) or GraphPad Prism (version 9.20).

## Results

14 eligible studies (Table 1) were included with 1663 inoculations performed, involving 1416 participants (median age 21 [IQR 20–25], males = 568). Of these studies, 6 were RCTs (4 double-blinded, 2 partially-blinded) and 8 were interventional cohort studies. S1 Fig describes the study timelines. Two previously described inoculation methods were used: 1) 'Pipette-to-nose' technique [17] (100µl challenge inoculum directly pipetted into each nare, n = 1308) 2) 'Hand-to-nose' technique [14] (participant transfers challenge inoculum pipetted onto the dorsum of their hand to their nose under supervision, utilising a higher dose, n = 355). Across all studies pneumococcal colonisation was defined as the detection of experimental pneumococcus in nasal wash samples at any timepoint post-inoculation by classical microbiology, with latex agglutination testing for serotype confirmation.

Two studies were identified but not included in the full analysis: 1) 'Feasibility of Experimental Human Nasal Colonisation' (REC ID 08/H1001/52, n = 20) 2) 'Safety, Tolerability, and Efficacy Study of Prophylactic S. Pneumoniae Vaccine Following Challenge With *S. Pneumoniae*'[20] (REC ID 14/NW/0355, n = 100) as complete data was not obtainable. From the data available, no pneumococcal-related SAEs were reported in these studies.

214 reported safety events were reviewed in detail (182 potentially pneumococcal related). Four challenge isolates were utilised: SPN6B, SPN3, SPN15B, and SPN23F. Four studies involved ≥1 inoculation, and one study involved three inoculations (Fig 2).

### No pneumococcal-related SAEs occurred post-experimental challenge

3 SAEs (3 individuals, 2 inoculated with SPN6B, 1 pre-inoculation, 0.21% of participants) were identified, all independently deemed unrelated to pneumococcal inoculation. All were significant medical events leading to hospitalisation and were considered SAEs. One participant (experimental SPN6B colonised) was admitted with tonsilitis for 48 hours. As non-toxigenic *Corynebacterum diptheriae* was cultured on throat swabs, this SAE was deemed unrelated to inoculation. The participant made a full, uneventful recovery following intravenous, then oral, antibiotics.

One further participant (experimental SPN3 colonised) reported an adverse event of special interest that was initially classed as an SAE. This individual developed otitis media with effusion following inoculation, progressing to tympanic perforation. A swab confirmed experimental SPN3 in aural discharge. They made a full recovery without sequelae, and following a detailed discussion with the DSMC this was downgraded to an AESI.

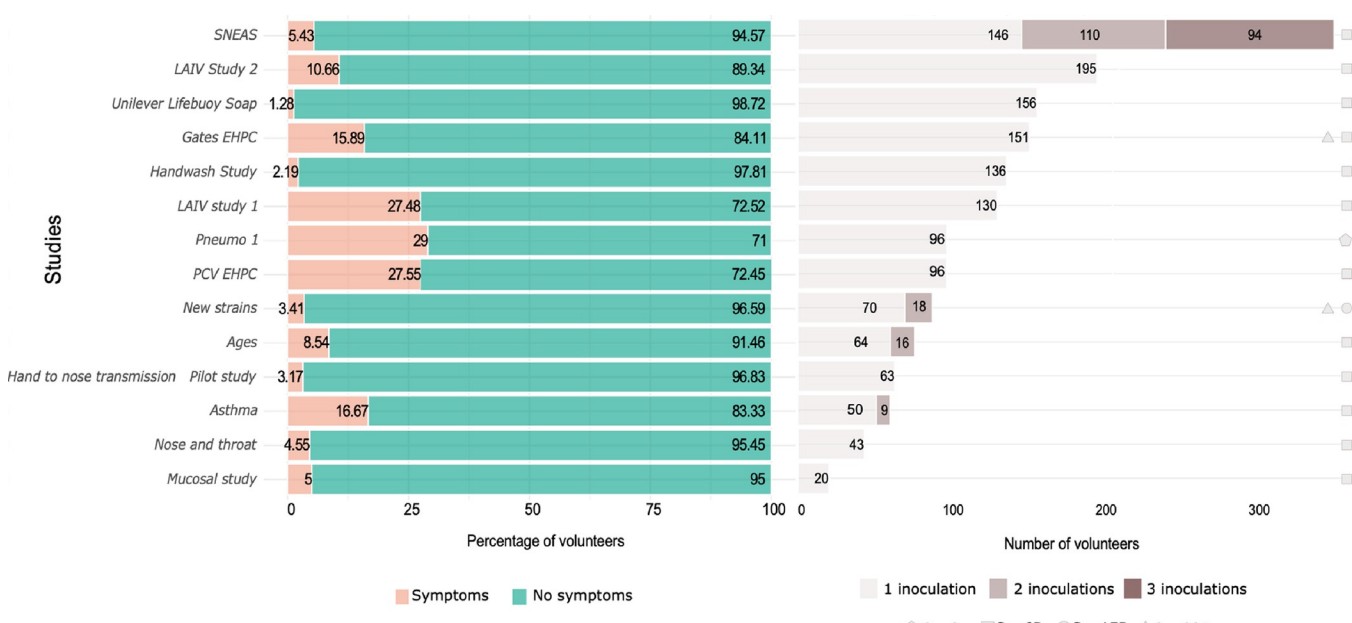

**Fig 2. Study size, number of inoculations per study and frequency of safety review for potential pneumococcal symptoms.** The stacked bar graph demonstrates the percentage of participants reporting symptoms in each study (pink-symptoms, green- no symptoms). The bar chart demonstrates the number of inoculations performed in each study. Most EHPC studies involved one inoculation per participant, 4 studies involved two or more inoculations per participant and one study involved three inoculations per participant.

## Most of the reported symptoms post-inoculation were mild

The majority of safety review events involved grade 1 (mild) symptoms (colonised group = 72.7% [120/165], non-colonised = 86.7% [124/143]). When the highest-grade symptoms reported per individual were compared, significantly more colonised participants reported mild (grade 1) symptoms (colonised = 79/658, non-colonised = 73/1005, $P$ = 0.001, OR 1.74, 95% CI 1.24–2.44), but there was no difference in grade 2 (moderate) symptoms (colonised = 17/658, non-colonised = 13/1005, $P$ = 0.06, OR = 2.02, 95% CI 1.02–4.12). Median symptom onset time was 5 days for colonised (IQR 2–9) compared to 3 (IQR 2–9) for non-colonised (Fig 3). No significant difference in onset time was observed, with the majority occurring <7 days. Colonised individuals were significantly more likely to have viral pathogens identified from throat swabs (viral swab positive total = 11/60 swabs, colonised = 9/11, non-colonised = 2/11, $P$ = 0.009, OR 7.10, 95% CI 1.77–32.88).

Most safety reviews were performed at routine visits (74.2% [135/182]) indicating symptoms were likely to be reported only when directly asked at pre-planned appointments. Significantly more extra assessments were performed in colonised individuals (colonised = 69/658, non-colonised = 66/1005, $P$ = 0.006, OR 1.67, 95% CI 1.16–2.35).

## Safety review events occurred more frequently in pneumococcal colonised participants

Using the one-stage individual patient data meta-analysis approach the overall unadjusted association between pneumococcal colonisation and safety review (for any symptoms) was OR = 1.67 (95% CI 1.22–2.29). When adjusted to account for important co-variates (age, gender, inoculum, inoculum method) the OR is 1.49 (95% CI 1.08–2.06). The two-stage meta-analysis (covariate unadjusted) provides similar estimates (Fig 4), with a meta-analytic OR of

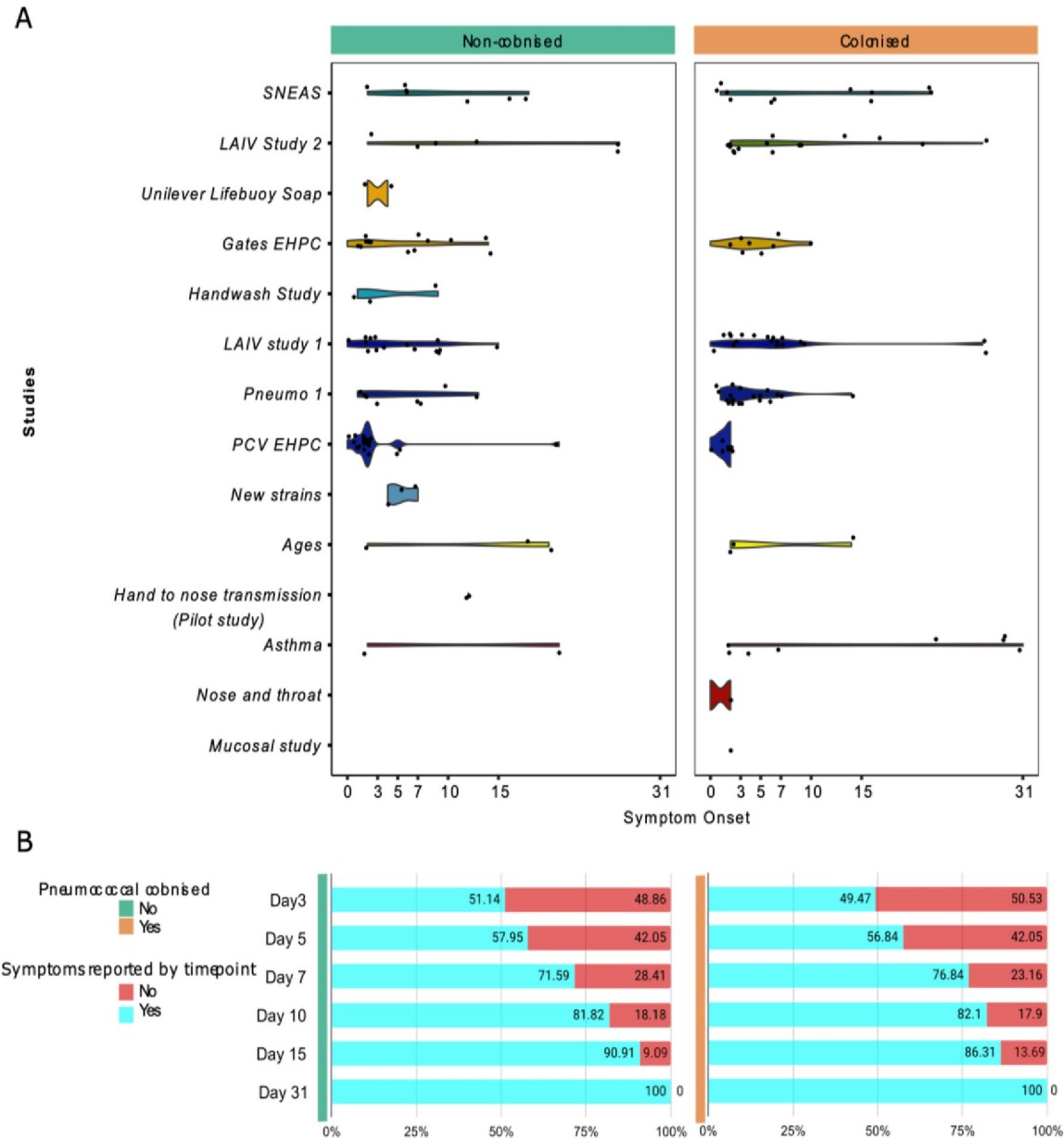

**Fig 3. Comparison of reported symptom onset time in colonised and non-colonised individuals.** A) Violin plots demonstrating the onset time for the first reported symptom following inoculation for colonised (orange) and non-colonised individuals (green). Each study is identified in a different colour. B) Stacked bar graph demonstrating the percentage of reported symptoms of the total that have occurred by each timepoint (days post-inoculation) in those colonised and non-colonised.

1.64 (95% CI 1.06–2.54). There is variation between studies, $I^2 = 35\%$ ($P = 0.09$), suggesting 35% of the heterogeneity in the association between pneumococcal colonisation and symptoms is attributable to inter-study heterogeneity.

The same analyses carried out using only potentially pneumococcal-related symptoms demonstrate safety review events occurred more frequently in colonised individuals (96/658 colonised vs 86/1005 non-colonised) with a one-stage individual patient data model estimated OR 1.81 (95% CI 1.28–2.56, $P = <0.001$), S3 Table. We hypothesised study season would impact safety review frequency for potential pneumococcal symptoms, with increased URT

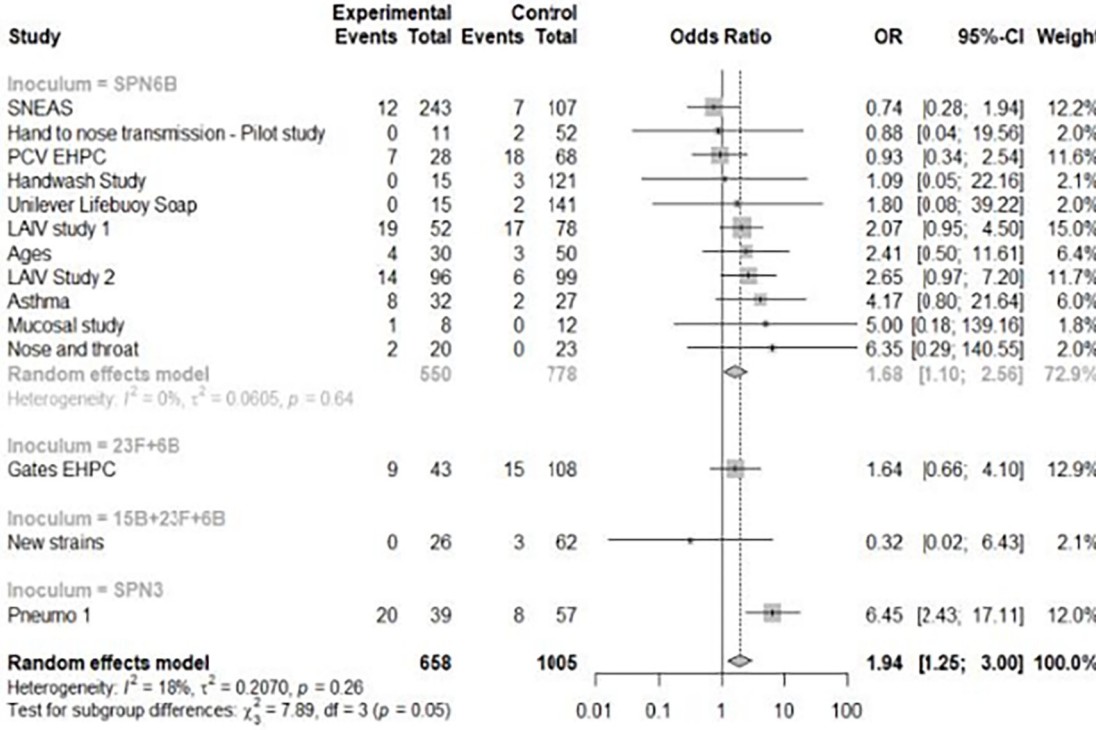

**Fig 4. Forest plots illustrating the OR for safety review post-inoculation for those pneumococcal colonised and non-pneumococcal colonised by study.** A) Forest plot of all safety reviews (combining potentially pneumococcal and non-pneumococcal related) for all included EHPC studies using an unadjusted random effects model. Effect sizes are reported as OR with 95% CI. Studies are ranked by their OR. The overall OR is shown in bold, and a dotted line is included for comparison across studies. B) Forest plot of all potentially pneumococcal safety reviews for all EHPC studies using an unadjusted random effects model, adjusted for inoculum serotype (SPN6B, SPN23F+SPN6B, SPN15B/SPN23F +SPN6B, SPN3).

symptoms over autumn/winter; however, the season was not associated with increased odds of review in the adjusted model. The participant's age and sex were also non-significant (age OR = 1.00 [95% CI 0.98–1.01 $P$ = 0.60], male sex OR = 0.72 [95% CI 0.51–1.01, $P$ = 0.063]), $I^2$ = 18% ($P$ = 0.26).

## Antibiotic initiation for safety reasons was low

1.62% of all participants (23/1416) commenced antibiotics for potential pneumococcal symptoms, with the majority colonised (colonised = 16/658, non-colonised = 7/1005, $P$ = 0.004, OR 3.55, 95% CI 1.54–9.23). 25% (4/16) of colonised participants requiring antibiotics were subsequently swab-positive for an additional potential pathogen (Rhinovirus, Influenza B, Respiratory Syncytial Virus (RSV) and non-toxigenic *Corynebacterium diptheriae*). The median time to commencing antibiotics was similar (colonised = 7 days [IQR 3–7.25], non-colonised = 8 days [IQR 4.5–11.5]). Sore throat and otalgia were the most frequent symptoms requiring antibiotics; both occurred more in colonised individuals (sore throat: colonised = 11/658, non-colonised = 3/1005, $P$ = 0.004, OR 5.68, 95% CI 1.77–19.10, otalgia: colonised = 5/658, non-colonised = 1/1005, $P$ = 0.04, OR 7.69, 95% CI 1.07–90.70).

## Reported potential pneumococcal symptoms were similar in both the pneumococcal colonised and non-colonised

While potential pneumococcal symptoms were reported more frequently in colonised individuals, symptom frequency and type were similar across all studies, as shown in S2 Fig. Sore throat, cough, and otalgia were the symptoms reported most often for both groups; however only sore throat (colonised = 57/658, non-colonised = 48/1005, $P$ = 0.002, OR 1.89, 95% CI 1.28–2.83) and malaise (colonised = 8/658, non-colonised = 2/1005, $P$ = 0.02, OR 6.18, 95% CI 1.53–29.0) occurred more frequently in colonised individuals (Fig 5A).

## Inoculum serotype influences symptomology with SPN3 inoculation associated with a higher frequency of reported safety events, grade of symptoms and requirement for antibiotics

Previous research has identified increased symptoms in those experimentally colonised with Spn3 [11] and this is driving the majority of the differences seen in colonised individuals. Forest plots from the two-stage meta-analysis (utilising a random effects model) with frequency of reported potential pneumococcal symptoms as outcome and stratified by inoculum serotype are presented (Fig 4B). The SPN3 group had a significantly higher OR for reported symptoms compared to SPN6B (SPN6B OR 1.68, 95% CI 1.10–2.56 vs SPN3 OR 6.45, 95% CI 2.43–17.11) and 23F (SPN23F OR 1.64, 95% CI 0.66–4.10).

When SPN3 was excluded from the symptom grading analysis, a higher proportion of symptoms were mild (grade 1) for colonised and non-colonised (grade 1 colonised = 77.0% [107/139], non-colonised = 93.4% [114/122]) but no statistically significant difference in grade 2 symptoms were observed (grade 2 colonised = 9/619, non-colonised = 12/948, $P$ = 0.82, OR = 1.15, 95% CI 0.46–2.83). SPN3 inoculation also drove antibiotic use, with reduced antibiotic requirement (0.90% of participants, [14/1567]) and no significant difference between colonised and non-colonised (colonised = 8/619, non-colonised = 4/948, $P$ = 0.18) following SPN3 exclusion.

Similarly, SPN3 colonisation influenced symptomology. When excluded from the analysis of the presenting symptoms (Fig 5B) only 'malaise' occurred more frequently in the colonised (colonised = 8/619, non-colonised = 2/948, $P$ = 0.018, OR 6.17, 95% CI 1.53–28.97). The same was seen when symptoms leading to antibiotics were considered, with no significant difference observed for the remaining serotypes.

Another critical variable is the inoculation method. This variable had less influence on review frequency, despite the 'Hand-to-nose' method utilising a $\log_{10}$-higher inoculum dose for SPN6B. When the one-stage meta-analysis approach was stratified by inoculation method,

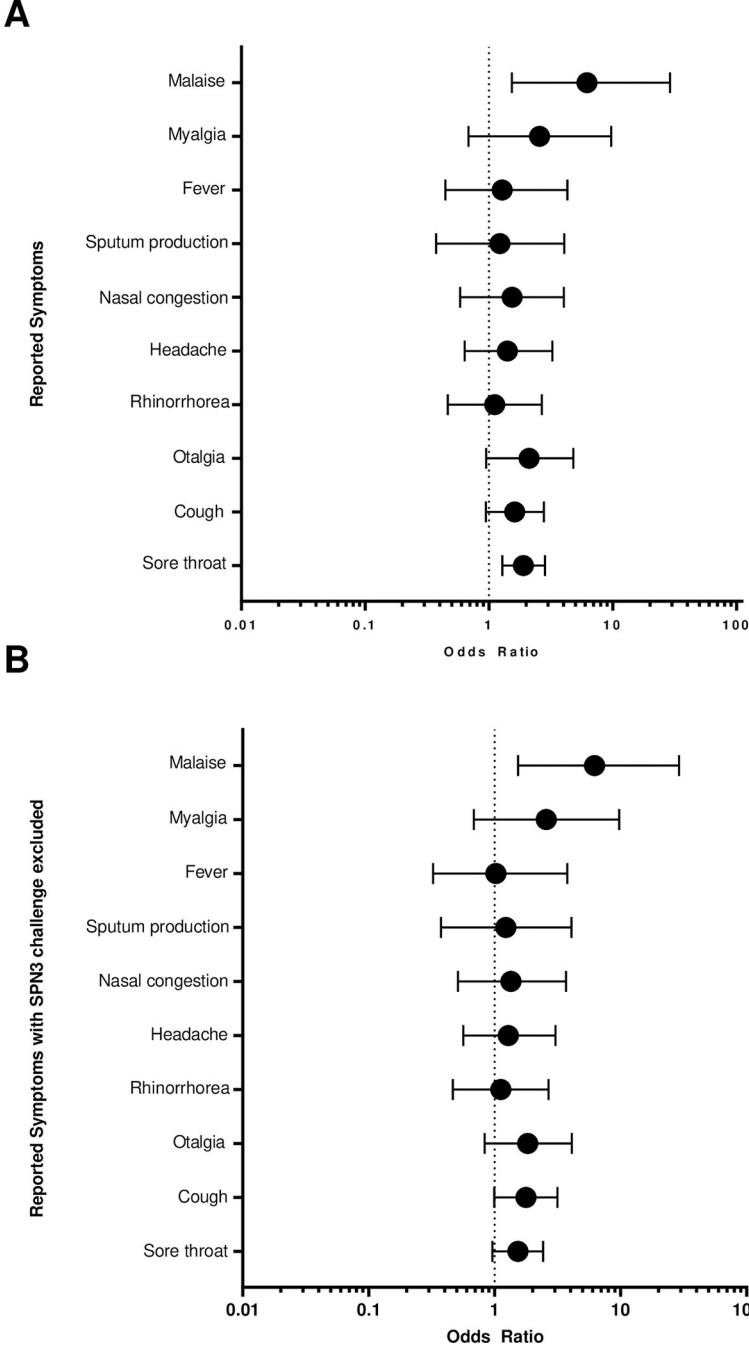

**Fig 5. Odds ratio plot of the most frequently reported symptoms in keeping with pneumococcal disease reported at safety review.** A) OR (95% CI) of each potentially pneumococcal symptom occurring in experimentally colonised individuals compared to non-colonised post-inoculation B) OR (95% CI) of each potential pneumococcal symptom occurring in experimentally colonised individuals compared to non-colonised, excluding SPN3 inoculated participants. Potential pneumococcal symptoms were pre-identified through a literature search (results shown in **S3 Table**). OR were calculated using Fisher's Exact Test. A dotted line identifies an OR of 1 for comparison between symptoms. Only 'sore throat' and 'malaise' are statistically significant, with only 'malaise' significant when SPN3 inoculated individuals are removed.

there was a small increase in review frequency for the pipette-to-nose group compared to the hand-to-nose group (pipette-to-nose OR = 2.0 [95% CI 1.18–3.38] vs hand-to-nose OR = 1.20 [95% CI 0.48–2.97], S3 Fig).

Three EHPC studies were CTIMPs (Table 1). Due to the increased regulatory requirements of CTIMPs, we hypothesised these would have a higher frequency of safety review episodes compared to non-CTIMPs. However, when adjusted for age, sex and inoculum received, CTIMPs had a similar OR for reporting pneumococcal-related symptoms as non-CTIMPs (CTIMPs OR 1.71, 95% CI 1.03–2.85, $P = 0.038$, Non-CTIMPs OR 2.02, 95% CI 1.28–3.19, $P = 0.0003$).

## The frequency of non-pneumococcal symptoms was similar in pneumococcal colonised and non-colonised individuals

No significant difference in non-pneumococcal symptoms was observed between colonised and non-colonised (colonised 11/658 vs non-colonised 21/1005, $P = 0.59$, OR = 0.80, 95% CI 0.38–1.64). This held across all serotypes and inoculation methods. The type of safety review (at vs outside of routine appointment) was not significantly different. 7 participants commenced antibiotics for non-pneumococcal symptoms, with similar rates in colonised and non-colonised groups (colonised = 2/658, non-colonised = 5/1005, $P = 0.71$). The reported non-pneumococcal symptoms were diverse, with no identifiable symptomology trend.

## Discussion

We have demonstrated that outpatient EHPC can be performed safely with appropriate screening and monitoring procedures as described. Symptoms requiring safety review are predominantly mild, infrequently require antibiotics, and their frequency is influenced by inoculum serotype. These data support the expanded use of the model to investigate additional serotypes, including adults at higher risk of pneumococcal disease (the primary target for improved pneumococcal vaccine development). Symptom review by research clinicians was influenced by knowledge of colonisation status, which likely affected clinical decision-making. Outpatient management is appropriate, but future risk assessments should consider inoculum serotype and maintain access to research-independent clinicians for colonisation-blinded review if severe symptoms develop. Fig 6 summarises the recommended features of a common EHPC protocol to ensure participant safety.

All three SAEs identified in this analysis were deemed unrelated to experimental inoculation. The frequency of symptoms requiring review was low, with symptom onset predominantly <7 days, supporting the continued use of an outpatient-based model. While there was variability in the frequency of reviews performed, only 12.92% of participants reported symptoms in keeping with pneumococcal disease. The majority were mild, only requiring assessment at routine appointments.

Low levels of heterogeneity were detected between studies, which was anticipated as all reported results are from one centre. However, the lack of a standardised severity grading measures limits the ability to compare the frequency and grade of adverse events occurring in individual CHIM programmes internationally. We suggest a standardised reporting methodology for all adverse events could allow wider comparisons between CHIM pathogens and ensure continued public confidence in CHIM methodology.

Pneumococcal colonised participants were reviewed more often than non-colonised (both for symptoms in keeping with pneumococcal disease and all-cause symptoms). This is likely due to the clinical team being aware of colonisation status during review, leading to extra vigilance for those colonised. We recommend blinding clinical staff to colonisation status where

| Pre-inoculation | Inoculation | Post-inoculation |
|---|---|---|
| • Inclusion and exclusion criteria as stated in S1 Table<br>• Screening blood tests with a minimum of Full blood count<br>• Physical examination and relevant medical history<br>• Establish if pneumococcal colonisation at baseline | • Antibiotic sensitive and serotype verified inoculum confirmed by an independent laboratory<br>• Ensure participant well<br>• 24 hour call physician contact details<br>• Access to inpatient care if necessary<br>• Participant education on symptoms of pneumococcal disease<br>• Clinician blinding to inoculum where feasible<br>• Rescue antibiotics | • Study information card<br>• Remote temperature monitoring for a minimum of 3 days<br>• Safety review by clinical team if symptoms with viral and bacterial throat swabs if clinically indicated<br>• Follow up for minimum of 10 days or until colonisation termination<br>• Termination of colonisation at last visit or if concerning symptoms |

**Fig 6. Summary of a common protocol features for safely conducting EHPC in an outpatient setting.** A schematic demonstrating the key features of a common protocol for safely performing EHPC pre- and post-pneumococcal inoculation.

practical, ensuring all participants are managed objectively. However, the inoculum serotype utilised significantly impacts symptom frequency, with SPN3 associated with significantly more frequent and specific symptoms than other serotypes. When SPN3 is excluded, the remaining symptoms are predominantly mild, and antibiotic use is reducing substantially. This finding suggests SPN3 diverges from an asymptomatic colonisation model to one of controlled mild infection. While this is a significant change for EHPC, it is in keeping with other CHIM, such as the Group A Streptococcus [21] challenge, whose primary outcome is pharyngitis rather than colonisation. Future SPN3 studies should inform participants of the increased frequency and nature of these symptoms.

URT symptoms are frequently seen in respiratory viral infection, which was identified more often in colonised individuals, and can influence pneumococcal carriage [22,23]. As this analysis focused on safety, it is unclear from our data whether reported symptoms are occurring primarily due to pneumococcal colonisation, the development of infection or a symptomatic viral infection occurring (but not identified) co-incident with asymptomatic pneumococcal colonisation.

Identification of safety events is reliant on accurate record-keeping. As discussed there is potential for bias in reporting and management of symptomatic participants if colonisation status is known before review. The study is limited to a single centre and it is, therefore, unclear from the data whether these findings can be extrapolated to other centres. Two studies

had incomplete safety records available. Retrospective symptom grading is subjective, even when performed by experienced medical teams. The medical team overseeing EHPC has evolved over the 10-year duration and there is likely to be some variability in clinical assessment. These limitations reduce the validity of conclusions regarding the frequency and type of symptoms occurring post-inoculation. However, this analysis provides a significant pooled population sample of a unique challenge methodology and applies robust statistical modelling. Importantly, the frequency of SAEs and overall antibiotic use is collated objectively across studies.

## Conclusion

These data demonstrate an outpatient-based pneumococcal CHIM utilising varied challenge isolates, inoculation methods, and participants can be conducted safely with appropriate monitoring (common protocol described in Fig 6). Antibiotic use was low and safety reviews were infrequent, irrespective of colonisation status. SPN3 had a higher frequency and grade of symptoms, but this risk can be safely mitigated with close monitoring. Our data support the continued development of EHPC within higher- risk populations, and the development of similar outpatient challenge models for vaccine research. This study represents the experience of EHPC in a UK regulatory process and adjustments may be required when utilised elsewhere. This model has recently been transferred to a low-and-middle-income country (LMIC) [24], and these data suggest that an -adapted safety framework based on that described here, could also allow the transfer of an SPN3 model to this setting. Improved standardisation of adverse event reporting in CHIM would facilitate a direct comparison of safety between challenge pathogens.

## Supporting information

**S1 Fig. Timeline of EHPC studies 2011–2021.** Gantt chart ranging from 2011 to 2022 with each EHPC study represented by a separate colour. Study commencement was defined as the time of first participant consent and the study end was defined as the last participant visit. (DOCX)

**S2 Fig. Heatmap of reported potential pneumococcal symptoms in EHPC studies.** All included studies are shown in different colours with each individual reporting symptoms on a separate row. All reported symptoms are shown in individual columns. Grade 1 severity symptoms shown in grey and grade 2 in black. The pneumococcal colonised are in green and non-pneumococcal colonised in blue. Each inoculum serotype is shown in different colours (SPN6-green, SPN3- pink, SPN15B- dark green, SPN23F-blue). Sore throat is the most commonly reported symptom. (DOCX)

**S3 Fig. Forest plot of all potentially pneumococcal related safety reviews across all included EHPC studies using a random effects model adjusting for inoculation method (pipette-to-nose method or hand-to-nose method).** IM = 1 refers to the' pipette- to-nose' method of inoculation and IM = 2 refers to the 'hand-to-nose method of inoculation. Effect sizes are reported as OR with 95% CI. The overall OR is shown in bold, and a dotted line included for comparison across studies. (DOCX)

**S1 Table. Inclusion and Exclusion criteria in Experimental Human Pneumococcal Challenge.** (DOCX)

**S2 Table. Search terms used to define pneumococcal symptoms.**
(DOCX)

**S3 Table. Potential Pneumococcal symptoms expected by the most frequent pneumococcal disease syndromes as defined by UK HSA.**
(DOCX)

**S4 Table. Grading score for adverse events.**
(DOCX)

**S1 File. Participant symptom reporting and safety review additional information.**
(DOCX)

**S2 File. Study data table.**
(XLSX)

## Author Contributions

**Conceptualization:** Ryan E. Robinson, Helen Hill, Wendi A. Shepherd, Daniella Mclenghan, Maia Lesosky, Stephen B. Gordon, Andrea M. Collins, Daniela M. Ferreira.

**Data curation:** Ryan E. Robinson, Wendi A. Shepherd, Patricia Gonzalez-Dias, Konstantinos Liatsikos, Samuel Latham, Fred Fyles, Klara Doherty, Phoebe Hazenberg, Fathimath Shiham, Daniella Mclenghan, Hugh Adler, Vicki Randles, Seher Zaidi, Angela Hyder-Wright, Elena Mitsi, Ben Morton, Stephen B. Gordon, Andrea M. Collins, Daniela M. Ferreira.

**Formal analysis:** Ryan E. Robinson, Christopher Myerscough, Nengjie He, Helen Hill, Patricia Gonzalez-Dias, Fred Fyles, Klara Doherty, Fathimath Shiham, Angela Hyder-Wright, Jamie Rylance, Maia Lesosky.

**Investigation:** Ryan E. Robinson, Konstantinos Liatsikos, Samuel Latham, Fathimath Shiham, Daniella Mclenghan, Elena Mitsi.

**Methodology:** Ryan E. Robinson, Christopher Myerscough, Nengjie He, Wendi A. Shepherd, Konstantinos Liatsikos, Samuel Latham, Fred Fyles, Klara Doherty, Phoebe Hazenberg, Hugh Adler, Vicki Randles, Seher Zaidi, Elena Mitsi, Ben Morton, Jamie Rylance, Maia Lesosky, Andrea M. Collins.

**Project administration:** Ryan E. Robinson, Christopher Myerscough.

**Resources:** Ryan E. Robinson.

**Supervision:** Elena Mitsi, Hassan Burhan, Maia Lesosky, Stephen B. Gordon, Andrea M. Collins, Daniela M. Ferreira.

**Validation:** Ryan E. Robinson, Christopher Myerscough, Patricia Gonzalez-Dias.

**Visualization:** Ryan E. Robinson, Christopher Myerscough, Patricia Gonzalez-Dias, Elena Mitsi, Maia Lesosky, Andrea M. Collins.

**Writing – original draft:** Ryan E. Robinson, Christopher Myerscough, Konstantinos Liatsikos.

**Writing – review & editing:** Ryan E. Robinson, Christopher Myerscough, Nengjie He, Helen Hill, Wendi A. Shepherd, Patricia Gonzalez-Dias, Konstantinos Liatsikos, Samuel Latham, Fred Fyles, Klara Doherty, Phoebe Hazenberg, Fathimath Shiham, Daniella Mclenghan, Hugh Adler, Vicki Randles, Seher Zaidi, Angela Hyder-Wright, Elena Mitsi, Hassan

Burhan, Ben Morton, Jamie Rylance, Maia Lesosky, Stephen B. Gordon, Andrea M. Collins, Daniela M. Ferreira.

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
