## [Decision Letter · Decision Letter 0]

3 Feb 2023

PONE-D-22-31116Comprehensive review of safety in Experimental Human Pneumococcal ChallengePLOS ONE

Dear Dr. Robinson,

Thank you for submitting your manuscript to PLOS ONE. After careful consideration, we feel that it has merit but does not fully meet PLOS ONE’s publication criteria as it currently stands. Therefore, we invite you to submit a revised version of the manuscript that addresses the points raised during the review process.

The reviewers recognized the importance of the work; however, we feel that several issues should be addressed, and the manuscript should be heavily edited before publication. 

We look forward to receiving your revised manuscript.

Kind regards,

Luis Felipe Reyes, M.D., Ph.D., MSc.

Academic Editor

PLOS ONE

Journal Requirements:

2. Please ensure you have included the registration number for the clinical trials referenced in the manuscript.

5. Please include a copy of Tables 2 and 3 which you refer to in your text on page 5, 6, 8 and 13.

Additional Editor Comments:

We have completed the revision of this manuscript. The reviewers recognized the importance of the work; however, we feel that several issues should be addressed, and the manuscript should be heavily edited before publication.

Reviewers' comments:

Reviewer's Responses to Questions

**Comments to the Author**

1. Is the manuscript technically sound, and do the data support the conclusions?

Reviewer #1: Partly

Reviewer #2: Yes

2. Has the statistical analysis been performed appropriately and rigorously? 

Reviewer #1: Yes

Reviewer #2: Yes

3. Have the authors made all data underlying the findings in their manuscript fully available?

Reviewer #1: No

Reviewer #2: Yes

4. Is the manuscript presented in an intelligible fashion and written in standard English?

Reviewer #1: No

Reviewer #2: Yes

5. Review Comments to the Author

Reviewer #1: This manuscript describes the UK experience with a human challenge model of pneumococcal colonization and its role in promoting vaccine efficacy trials. The model has been a very important and unique asset to clinical trials of pneumococcal interventions and this is a 10 year timely review of this remarkable experience. It would be most helpful to clarify some points so that it is clear to others how replicate this work based on what this data set shows to be a protocol with common features that has proven safety.

1) It is important to conclude that this study represents experience using UK regulatory process that may need to be adjusted if used elsewhere. While obvious, it is important to state.

2) There are many significant grammatical errors throughout

3) The utility of this study to other investigators depends on conveying how a set of studies that used similar but not identical protocols can define a common protocol that provides good outcome under safe conditions. It would be critical to conclude what this common protocol looks like.

4) As written it is not possible to determine the parameters of the various studies such as exclusion criteria, screening done before and during the studies, etc. Fig 1 refers to ODS Table 6 for much of this information that would critical to any investigator trying to use this model with the expectation of safety. Yet Fig ODS Table 6 doesn't exist in the Supp data.

5) Fig 4B is confusing. It appears that the studies reach 100% colonization and 100% have symptoms.

6) Fig 6 is very confusing with no apparent utility since the reader doesn't know what is the common protocol to all these various studies. Which of the many lines indicates the optimal study to use?

Reviewer #2: Dr. Robinson and colleagues present their manuscript, ‘Comprehensive review of safety in Experimental Human Pneumoccal Challenge’. The authors provide a single center review of all pneumococcal challenges over a 10-year period, for a cohort of 1400 individuals. Overall, they observed a very low rate of serious adverse events, and a limited number of cases that required use of antimicrobials. There appeared to be a higher burden of adverse events associated with serotype 3, but even these were limited in severity. Overall, authors suggest the risks for pneumococcal challenge are limited and manageable.

Strengths

Well-organized and written manuscript, with thorough breakdown of available data. Although limited to a single site, the authors report on numerous studies across a 10-year period. The assessment of safety reports appears to have been thorough and reasonably controlled for bias.

Major weaknesses

The major weakness is whether the observations from this single site analysis apply more broadly. The number of participants and inoculations, as well as the time span of accumulated studies are reassuring. The others do not comment on the diversity of study teams conducting the inoculation and symptom findings, that could be one additional approach to demonstrating consistency of observations.

The remainder of limitations are inherent to this work, where incidental viral infection or unrelated symptoms confound challenge-specific observations. Here, the authors handle the data and discussion appropriately.

Minor comments

Figure 2 could be moved to supplement without diminishing the manuscript.

Similarly, the impact of associations provided in figure 6 is limited. For participants that had recover of a viral pathogen, is there a distinct pattern of symptoms? Without further interpretation, I think there is little benefit to including.

6. PLOS authors have the option to publish the peer review history of their article (what does this mean?). If published, this will include your full peer review and any attached files.

Reviewer #1: No

Reviewer #2: **Yes: **Nathaniel Erdmann

---

## [Author Response · Author response to Decision Letter 0]

20 Mar 2023

Journal Requirements

C1) Please ensure that your manuscript meets PLOS ONE's style requirements, including those for file naming. The PLOS ONE style templates can be found at

R1) The article has now been updated to meet the PLOS one style requirements as requested.

C2) Please ensure you have included the registration number for the clinical trials referenced in the manuscript.

R2) All registration numbers for the included clinical trials are stated in Table 1. 

C3) Please include your full ethics statement in the ‘Methods’ section of your manuscript file. In your statement, please include the full name of the IRB or ethics committee who approved or waived your study, as well as whether or not you obtained informed written or verbal consent. If consent was waived for your study, please include this information in your statement as well.

R3) This is now stated in the methods section as requested, “All included studies had approval from a UK Research Ethics Committee (NHS Health research Authority REC North West – Liverpool Central, REC IDs included in Table 1) and had been prospectively registered on a clinical trials database where required. Prospective approval for this service evaluation was obtained from the Liverpool University Hospitals Trust Audit Department (ID 10354). All participants provided written informed consent”

C4) In your Data Availability statement, you have not specified where the minimal data set underlying the results described in your manuscript can be found. PLOS defines a study's minimal data set as the underlying data used to reach the conclusions drawn in the manuscript and any additional data required to replicate the reported study findings in their entirety. All PLOS journals require that the minimal data set be made fully available. For more information about our data policy, please see http://journals.plos.org/plosone/s/data-availability.

R4) The raw data has anonymised and included as a supporting information file. 

C5) Please include a copy of Tables 2 and 3 which you refer to in your text on page 5, 6, 8 and 13.

R5) These tables were provided in the supplementary materials. They have been re-labelled in the text using the PLOS ONE style guide to read S1 table, S2 Table, S3 Table for clarity. 

Additional Editor Comments:

We have completed the revision of this manuscript. The reviewers recognized the importance of the work; however, we feel that several issues should be addressed, and the manuscript should be heavily edited before publication.

Reviewers' comments:

C6) Is the manuscript technically sound, and do the data support the conclusions?

Reviewer #1: Partly

Reviewer #2: Yes

R6) Thank you for this feedback

C7) Has the statistical analysis been performed appropriately and rigorously? 

Reviewer #1: Yes

Reviewer #2: Yes

R7) Thank you for this feedback

C8) Have the authors made all data underlying the findings in their manuscript fully available?

Reviewer #1: No

Reviewer #2: Yes

R8) The data was going to be unloaded to a repository on publication but has now been provided as a supporting information in an anonymised manner to protect participants confidentiality. 

C9) Is the manuscript presented in an intelligible fashion and written in standard English?

Reviewer #1: No

Reviewer #2: Yes

R9) Thank you for this feedback, we have made alterations to the text to take this into account. 

Review Comments to the Author

Reviewer #1: 

This manuscript describes the UK experience with a human challenge model of pneumococcal colonization and its role in promoting vaccine efficacy trials. The model has been a very important and unique asset to clinical trials of pneumococcal interventions and this is a 10 year timely review of this remarkable experience. It would be most helpful to clarify some points so that it is clear to others how replicate this work based on what this data set shows to be a protocol with common features that has proven safety.

C10) It is important to conclude that this study represents experience using UK regulatory process that may need to be adjusted if used elsewhere. While obvious, it is important to state.

R10) Thank you for this feedback, this has now been added to the conclusion; “This study represents the experience of EHPC in a UK regulatory process, and adjustments may be required when utilised elsewhere”. 

C11) There are many significant grammatical errors throughout

R11) Significant effort has been made to address this throughout the text.

C12) The utility of this study to other investigators depends on conveying how a set of studies that used similar but not identical protocols can define a common protocol that provides good outcome under safe conditions. It would be critical to conclude what this common protocol looks like.

R12) Thank for this feedback. We agree that further description of the common study design and protocol features is an important element of the study. In order to address this, we have now included two new figures and reference to them in the text. The first (Fig 1A) further describes the normal EHPC study format. The second (Fig 6) describes the key features of a common EHPC protocol that can provide a safe participant outcome. 

A new table (S1 table) describing the inclusion and exclusion criteria utilised across all EHPC studies has been included to assist EHPC development at new centres. These changes have been reflected in the text. In addition, we have included reference to these safety features observed in the common protocol in the conclusion as suggested. 

C13) As written it is not possible to determine the parameters of the various studies such as exclusion criteria, screening done before and during the studies, etc. Fig 1 refers to ODS Table 6 for much of this information that would critical to any investigator trying to use this model with the expectation of safety. Yet Fig ODS Table 6 doesn't exist in the Supp data.

R13) Thank you for this feedback, we have now included a table of the common inclusion and exclusion criteria across all EHPC studies (S1 Table). The safety monitoring and review process has been expanded in the methods to address this. 

C14) Fig 4B is confusing. It appears that the studies reach 100% colonization and 100% have symptoms.

R14) Thank you for this feedback. Figure 4B (now Fig 3B) illustrates the percentage of symptomatic participants who had reported their symptoms by each sequential timepoint. We have made this clearer now in the figure and the figure legend. 

C15) Fig 6 is very confusing with no apparent utility since the reader doesn't know what is the common protocol to all these various studies. Which of the many lines indicates the optimal study to use?

R16) We have now removed this figure and reference to it in the text. 

Reviewer #2: 

Dr. Robinson and colleagues present their manuscript, ‘Comprehensive review of safety in Experimental Human Pneumococcal Challenge’. The authors provide a single centre review of all pneumococcal challenges over a 10-year period, for a cohort of 1400 individuals. Overall, they observed a very low rate of serious adverse events, and a limited number of cases that required use of antimicrobials. There appeared to be a higher burden of adverse events associated with serotype 3, but even these were limited in severity. Overall, authors suggest the risks for pneumococcal challenge are limited and manageable.

Strengths

Well-organized and written manuscript, with thorough breakdown of available data. Although limited to a single site, the authors report on numerous studies across a 10-year period. The assessment of safety reports appears to have been thorough and reasonably controlled for bias.

Major weaknesses

C17) The major weakness is whether the observations from this single site analysis apply more broadly. 

R17) Thank you, we agree this is a limitation of the work and has now been highlighted in the paper. 

C18) The number of participants and inoculations, as well as the time span of accumulated studies are reassuring. The others do not comment on the diversity of study teams conducting the inoculation and symptom findings, that could be one additional approach to demonstrating consistency of observations.

R18) Thank you for this feedback. We have added this as a limitation in the discussion. ‘The medical team overseeing EHPC has evolved over the 10-year duration and there is likely to be some variability in clinical assessment’. Data regarding the individual inoculator was not available for assessment but could be studied in follow up studies. 

C19)The remainder of limitations are inherent to this work, where incidental viral infection or unrelated symptoms confound challenge-specific observations. Here, the authors handle the data and discussion appropriately.

R19) Thank you for your feedback. 

Minor comments

C20) Figure 2 could be moved to supplement without diminishing the manuscript.

R20) Thank you for this feedback, we have moved this figure to the supplement as suggested. 

C21) Similarly, the impact of associations provided in figure 6 is limited. For participants that had recover of a viral pathogen, is there a distinct pattern of symptoms? Without further interpretation, I think there is little benefit to including.

R21) This figure and reference to it has now been removed from the text.

---

## [Decision Letter · Decision Letter 1]

29 Mar 2023

Comprehensive review of safety in Experimental Human Pneumococcal Challenge

PONE-D-22-31116R1

Dear Dr. Robinson,

We’re pleased to inform you that your manuscript has been judged scientifically suitable for publication and will be formally accepted for publication once it meets all outstanding technical requirements.

Kind regards,

Luis Felipe Reyes, M.D., Ph.D., MSc.

Academic Editor

PLOS ONE

Reviewers' comments:

Reviewer's Responses to Questions

**Comments to the Author**

1. If the authors have adequately addressed your comments raised in a previous round of review and you feel that this manuscript is now acceptable for publication, you may indicate that here to bypass the “Comments to the Author” section, enter your conflict of interest statement in the “Confidential to Editor” section, and submit your "Accept" recommendation.

Reviewer #1: All comments have been addressed

2. Is the manuscript technically sound, and do the data support the conclusions?

Reviewer #1: Yes

3. Has the statistical analysis been performed appropriately and rigorously? 

Reviewer #1: Yes

4. Have the authors made all data underlying the findings in their manuscript fully available?

Reviewer #1: Yes

5. Is the manuscript presented in an intelligible fashion and written in standard English?

Reviewer #1: Yes

6. Review Comments to the Author

Reviewer #1: (No Response)

7. PLOS authors have the option to publish the peer review history of their article (what does this mean?). If published, this will include your full peer review and any attached files.

Reviewer #1: **Yes: **Elaine Tuomanen, MD

---

## [Editor Report · Acceptance letter]

14 Apr 2023

PONE-D-22-31116R1 

Comprehensive review of safety in Experimental Human Pneumococcal Challenge 

Dear Dr. Robinson:

I'm pleased to inform you that your manuscript has been deemed suitable for publication in PLOS ONE. Congratulations! Your manuscript is now with our production department. 

Kind regards, 

on behalf of

Dr. Luis Felipe Reyes 

Academic Editor

PLOS ONE